# Distinct cellular and molecular mechanisms for β3 adrenergic receptor-induced beige adipocyte formation

**Yuwei Jiang[1], Daniel C Berry[1†]\*, Jonathan M Graff[1,2]\***

[1]Division of Endocrinology, Department of Internal Medicine, University of Texas Southwestern Medical Center, Dallas, United States; [2]Department of Molecular Biology, University of Texas Southwestern Medical Center, Dallas, United States

**Abstract** Beige/brite adipocytes are induced within white adipose tissues (WAT) and, when activated, consume glucose and fatty acids to produce heat. Classically, two stimuli have been used to trigger a beiging response: cold temperatures and β3-adrenergic receptor (*Adrb3*) agonists. These two beiging triggers have been used interchangeably but whether these two stimuli may induce beiging differently at cellular and molecular levels remains unclear. Here, we found that cold-induced beige adipocyte formation requires *Adrb*1, not *Adrb3*, activation. *Adrb*1 activation stimulates WAT resident perivascular (*Acta2*+) cells to form cold-induced beige adipocytes. In contrast, *Adrb3* activation stimulates mature white adipocytes to convert into beige adipocytes. Necessity tests, using mature adipocyte-specific *Prdm16* deletion strategies, demonstrated that adipocytes are required and are predominant source to generate *Adrb3*-induced, but not cold-induced, beige adipocytes. Collectively, we identify that cold temperatures and *Adrb3* agonists activate distinct cellular populations that express different β-adrenergic receptors to induce beige adipogenesis.

DOI: https://doi.org/10.7554/eLife.30329.001

\*For correspondence:
dcb37@cornell.edu (DCB);
jon.graff@utsouthwestern.edu
(JMG)

Present address: †Division of Nutritional Sciences, Cornell University, Ithaca, United States

## Introduction

The ability to defend core body temperature in response to cold environments is, in part, mediated by non-shivering thermogenesis, through the recruitment and activation of brown and beige/brite (brown-in-white) adipose tissues (*Cannon and Nedergaard, 2004*). Beige adipocytes are activated in white adipose tissue (WAT) by cold exposure or by beta3-adrenergic receptor (*Adrb3*) agonists via the sympathetic nervous system (*Young et al., 1984*). Activation of beige adipocyte thermogenesis requires the induction of numerous thermogenic and mitochondrial genes such as uncoupling protein 1 (*Ucp1*) (*Cannon and Nedergaard, 2004*). To perform thermogenic function, beige adipocytes, like classical brown adipocytes, take up substantial amounts of glucose and free fatty acids and convert these molecules into heat rather than ATP. This uncoupling ability is clinically desirable because it has the potential to lower body fat percentage, reduce blood sugars and increase metabolic rate (*Sidossis and Kajimura, 2015*; *Blondin et al., 2014*; *Blondin et al., 2017*). Using radiolabeled glucose uptake and positron electron tomography (PET) imaging, brown/beige fat has been identified in cold-exposed adult humans (*Cypess et al., 2009*; *van Marken Lichtenbelt et al., 2009*). These imaging studies have indicated that glucose uptake occurs often in the supraclavicular region. Furthermore, supraclavicular WAT biopsies from cold-exposed humans have shown the presence and activation of brown/beige fat, which appears to resemble rodent beige fat rather than rodent brown fat (*Shinoda et al., 2015*; *Wu et al., 2012*). In addition to cold exposure, *Adrb3* agonists also stimulate and promote beige adipocyte formation and activation (*Himms-Hagen et al., 1994*). However, clinically used non-selective *Adrb3* agonists have led to off target side-effects such

**eLife digest** Excess accumulation of a type of fat called white fat is associated with obesity and metabolic problems. White fat cells store energy. White fat tissue also contains some beige fat cells, which burn fats and sugars to produce heat. Cold temperatures trigger the production and activity of beige fat cells, which allows the body to stay warm. People with obesity tend to have less beige fat and more white fat. This has led scientists to test whether treatments that increase the number of beige fat cells a person has could reduce fat mass and improve metabolism.

To develop treatments that increase beige fat, scientists must first understand where it comes from and how cold and other factors stimulate its growth. Recent studies have shown that smooth muscle cells, which surround blood vessel walls, make cold-induced beige fat cells. A widely used drug that turns on the β3 adrenergic receptor, which is found in the cell membrane, also boosts the creation of beige fat cells. Yet, it was not clear exactly how cold or this drug triggers the production of beige fat.

Now, Jiang et al. show that drugs that target β3 adrenergic receptors cause white fat cells in mice to change into beige fat cells. The experiments also showed that cold turns on a different receptor called the β1 adrenergic receptor on smooth muscle cells causing them to make beige fat cells. This shows that there is more than one source for beige fat cells in the body and that different strategies for increasing beige fat cell numbers do not work the same way.

More studies are needed to learn whether beige fat cells produced after exposure to cold or drugs behave in the same way and have similar affects on metabolism. This could help scientists determine if one of these strategies could make a better treatment for obesity or other metabolic disorders.

DOI: https://doi.org/10.7554/eLife.30329.002

as tachycardia and hypertension, which precludes the use of these *Adrb3* agonists as beiging agents in the clinic (*Arch, 2011*). Yet, recent efforts by Cypess and colleagues have identified a more selective *Adrb3* agonist, mirabergon (MB), which is clinically used to treat overactive bladder conditions, as a potential human 'beiger' (*Cypess et al., 2015*).

Numerous studies have proposed various origins of beige adipocytes. For example, the notion that white adipocytes can transdifferentiate into beige adipocytes has been a conventional notion (*Barbatelli et al., 2010*; *Smorlesi et al., 2012*). On the other hand, several recent perivascular genetic fate-mapping studies have indicated that cold-induced beige adipocytes are generated from blood vessel-derived progenitors within WAT. For instance, perivascular smooth muscle cells that express *Myh11*, *Pdgfrb* and *Acta2* have been shown to be a progenitor cell pool for cold-induced beige adipocytes (*Berry et al., 2016*; *Long et al., 2014*; *Shao et al., 2016*). Also, platelet-derived growth factor receptor α (*Pdgfra*) positive fibroblasts can form some *Adrb3*-induced beige adipocytes within the perigonadal adipose tissue (PGW) but not within the subcutaneous inguinal adipose (IGW) WAT depot (*Lee et al., 2012*). Yet, *Pdgfra+* cells do not appear to serve as a cold-induced beige progenitor pool (*Lee et al., 2015*; *Berry et al., 2016*). Thus, it is unclear if smooth muscle cells are the major contributing cellular source for *Adrb3*-induced beige adipocytes.

Mechanistically, studies have also suggested that both cold and *Adrb3* stimulate brown adipocyte activation and beige adipocyte recruitment similarly through the activation of *Adrb* (*Ramseyer and Granneman, 2016*). Norepinephrine signaling, via *Adrb3*, not only stimulates thermogenic action in mature brown adipocytes but also recruits and activates beige adipocytes within WAT (*Grujic et al., 1997*). Further, norepinephrine signaling through *Adrb3*-cAMP activation upregulates thermogenic genes such as *Ucp1* potentiating the beiging process (*Cao et al., 2001*). However, cellular expression studies have indicated that *Adrb3* expression is restricted to mature white and brown adipocytes and is not expressed within the stromal vascular fraction (SVF), the proposed site of both white and beige progenitors (*Collins et al., 1994*; *Berry et al., 2016*). In contrast to *Adrb3*, *Adrb*1 is expressed in the SVF, not in mature adipocytes, and is thought to mediate the proliferation and differentiation of classical brown adipocyte progenitors (*Bronnikov et al., 1992*). Thermogenically, *Adrb*1 null mice are unable to defend their body temperature in response to the cold (*Ueta et al., 2012*); whereas, transgenic overexpression of *Adrb*1 showed robust cold-induced beiging potential

within WAT (*Soloveva et al., 1997*). Overall, several studies suggest mechanistic differences between cold- and *Adrb3*-induced beige adipocyte formation (*Vosselman et al., 2012*). But whether these two receptors mediate or recruit beige adipocytes similarly remains unknown.

Here, we show that smooth muscle/mural cells (*Acta2* and *Myh11*) do not fate-map into *Adrb3*-induced beige adipocytes rather most *Adrb3*-induced beige adipocytes emanate from pre-existing white adipocytes. White adipocyte beiging necessity tests demonstrate the requirement of white adipocytes to form a significant percentage of beige adipocytes. Moreover, we find that that cold-induced beiging requires *Adrb1* activation but not *Adrb3* signaling. These data suggest that several cellular sources exist for beige adipocyte formation, which could provide mechanistic insight and clinical utility into enhancing beige adipocyte formation, function and perdurance.

## Results

### Difference between cold- and *Adrb3*-induced beiging

To begin to explore possible differences between cold-induced and *Adrb3*-induced beiging, we evaluated the relative appearance of in vivo beige adipogenesis. *C57BL/6J*-inbred mice were randomized to room temperature (23°C), cold (6.5°C) or CL316,243 (CL, 1 mg/Kg), a *Adrb3* selective agonist, for 1, 3, or 7 days. After 1 day of cold exposure, beige adipocyte appearance was minimal within the subcutaneous inguinal adipose depot (IGW), the predominant beiging WAT. Conversely, 1 day of CL treatment produced some *Ucp1+* multilocular beige adipocytes and after 3 days of CL many *Ucp1+* beige adipocytes could be observed. In contrast, after 3 days of cold exposure, a small number of *Ucp1+* beige adipocytes were present. By day 7, both beiging agents produced many *Ucp1+* beige adipocytes throughout the IGW depot (*Figure 1A* and *Figure 1—figure supplement 1*). These findings were also confirmed by quantitative real time-PCR analysis of beige and thermogenic gene expression (*Figure 1—figure supplement 1*).

To continue to test differences between cold- and *Adrb3*-induced beiging, we aged (six months) *C57BL/6J* inbred male mice. Aged mice have been shown to be defective in cold-induced beige adipocyte formation attributed to a cellular aging senescence-like phenotype of beige progenitors (*Rogers et al., 2012*; *Berry et al., 2017*). Mice were then randomized to room temperature, cold or CL for seven days. Cold exposure was unable to induce beiging or a thermogenic program, as previously reported (*Figure 1B–F*) (*Berry et al., 2017*). Strikingly, CL induced robust beiging, glucose consumption, and thermogenic gene expression compared to room temperature controls (*Figure 1B–G*).

### Smooth muscle cells do not generate *Adrb3*- induced beige adipocytes

To continue to explore potential difference between cold-induced and *Adrb3*-induced beige adipocytes, we attempted to probe whether perivascular mural cells that express *Acta2*, which function as a major cellular source for cold-induced beige adipocytes, also served as a progenitor source for *Adrb3* activation. Recent studies have used the *Acta2^Cre-ERT2*; *Rosa26R^RFP* as beige fate-mapping model (*Figure 2A*) (*Berry et al., 2016*; *Wendling et al., 2009*). Notably, neither the *Acta2*-driven reporter nor endogenous *Acta2* is expressed in beige or brown adipocytes or white adipocytes (*Berry et al., 2016*). To induce recombination, one dose of tamoxifen (TM, 50 mg/Kg) for two consecutive days was administered to two-month-old *Acta2^Cre-ERT2*; *Rosa26R^RFP* male mice (*Figure 2A*). After a two-week TM washout period, mice were randomized to vehicle, CL, or mirabegron (MB, 1 mg/Kg), a clinically used *Adrb3* agonist, for seven days (*Figure 2A*). Histological assessment revealed both CL and MB induced beige adipocyte formation, similarly, in the IGW and PGW WAT depots (*Figure 2—figure supplement 1*). Nevertheless, beige adipocytes induced by CL- or MB-did not originate from an *Acta2-RFP* perivascular source (*Figure 2B* and *Figure 2—figure supplement 2*). *Acta2-RFP* expression was restricted to the vasculature under both *Adrb3* agonist treatments, similar to vehicle sections (*Figure 2B* and *Figure 2—figure supplement 2*). We next tested if chronic cold or CL treatment could provoke *Acta2+* mural cells to form beige adipocytes. We cold exposed or CL treated TM-induced *Acta2-RFP* mice for 14 days. Fate mapping studies demonstrated that *Acta2+* cells formed beige adipocytes after 14 days of cold exposure, comparable to seven days of cold treatment (*Figure 2—figure supplement 3*) (*Berry et al., 2016*), However, CL-induced *Ucp1+* beige adipocytes were RFP negative after 14 days of treatment (*Figure 2—figure*

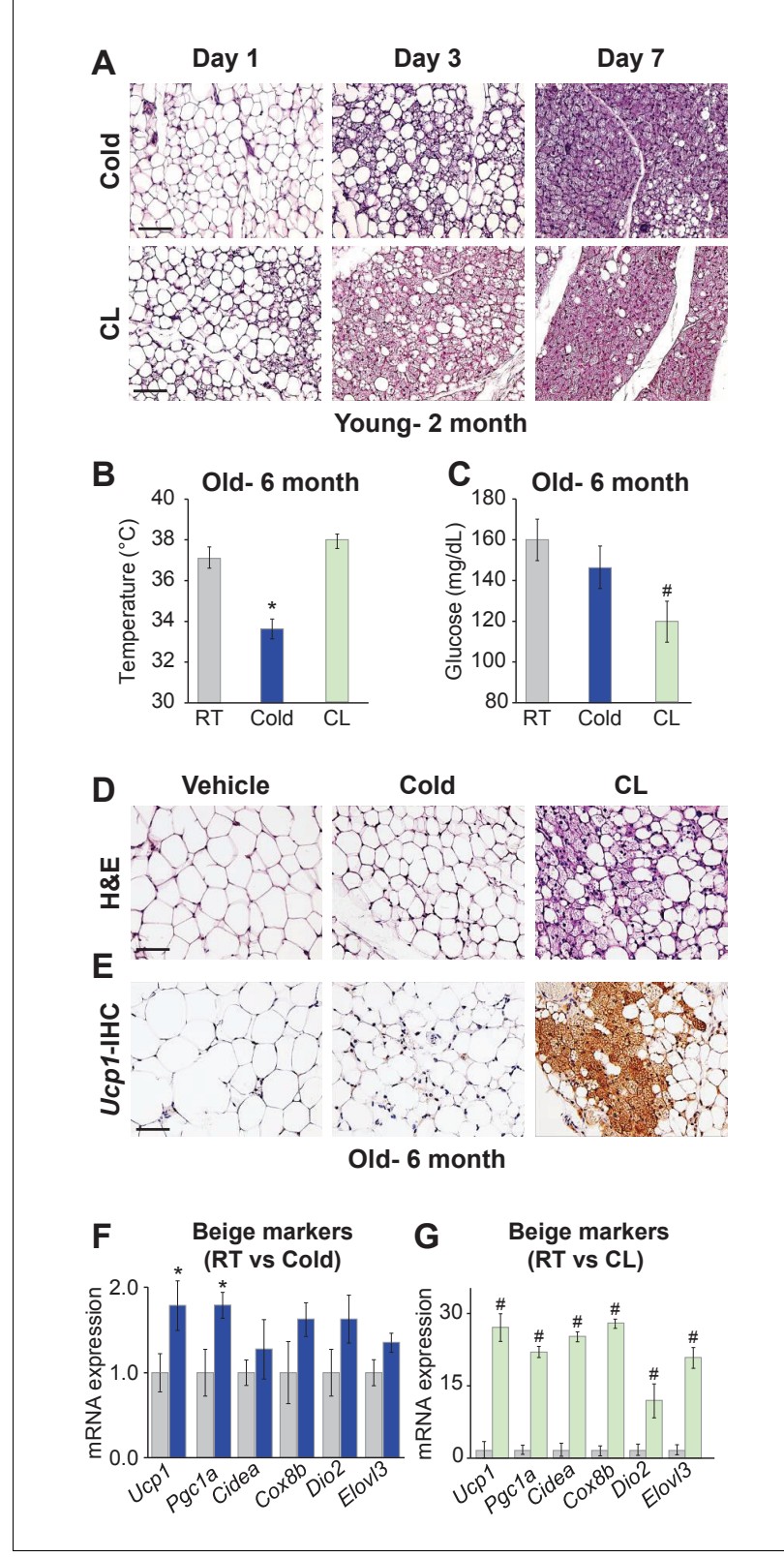

**Figure 1.** Differences between cold exposure and *Adrb3* agonists-induced beiging. (**A**) Representative H&E-stained images of sections from IGW depots from two-month-old male mice were cold exposed (6.5℃) or treated with CL (1 mg/Kg/mouse/day) for one, three or seven days (*n*= 6/group). (**B**) Rectal temperatures from six-month-old male mice maintained at room temperature, cold exposed or treated with CL for seven days (*n*= 10/group). (**C**)
*Figure 1 continued on next page*

*Figure 1 continued*

Sera glucose from mice described in (B). (D) Representative H&E-stained images of IGW depot sections from mice described in (B). (E) Representative *Ucp1* immunohistochemistry (IHC)-stained images of IGW depot sections from mice described in (B). (F) mRNA expression of beige and thermogenic genes from mice described in (B) maintained at room temperature or cold exposed for seven days. (G) mRNA expression of beige and thermogenic genes from mice described in (B) maintained at room temperature or treated with CL for seven days. *p<0.05 unpaired t-test, two tailed: Cold exposed compared to room temperature mice. #p<0.05 unpaired t-test, two tailed: CL-treated compared to vehicle room temperature-treated mice. Scale bar = 100 μm.

DOI: https://doi.org/10.7554/eLife.30329.003

The following figure supplement is available for figure 1:

**Figure supplement 1.** Differences between cold exposure and *Adrb3* agonists induced beiging.

DOI: https://doi.org/10.7554/eLife.30329.004

*supplement 3*). We then examined if age impacts *Acta2-RFP* fate mapping into *Adrb3*-induced beige adipocytes. We administered TM to P30, P90 and P180 *Acta2^Cre-ERT2*; *RFP* mice and then randomized mice to vehicle or CL for seven days, 2 weeks post TM. Histological assessment demonstrated that *Adrb3*-induced *Ucp1+* beige adipocytes were *Acta2-RFP* negative (*Figure 2—figure supplement 4*). Taken together, these data suggest that *Acta2+* mural cells do not serve as a progenitor cell source for *Adrb3*-induced beige adipocytes.

This negative tracing by *Acta2-RFP* into *Adrb3*-induced beige adipocytes may be, in part, due to undesired biological actions of TM on white adipose tissues thereby masking or altering fate mapping capabilities (*Ye et al., 2015*). To overcome possible TM-tracing issues, we employed a doxycycline (Dox, 0.5 mg/L) inducible *Acta2^rtTA*; *TRE-Cre*; *Rosa26R^RFP* mouse model (*Figure 2C*) (*Berry et al., 2016*). *Acta2^rtTA-RFP* was restricted to the vasculature and did not generate CL- or MB-induced beige adipocytes, confirming our TM-induced *Acta2^Cre-ERT2*; *Rosa26R^RFP* fate-mapping studies (*Figure 2D* and *Figure 2—figure supplement 5*).

To support our *Adrb3 Acta2* beiging fate-mapping studies, we performed two necessity strategies: (1) a cell-killing strategy (diphtheria toxin fragment A; DTA) (*Ivanova et al., 2005*) and (2) a blockade of adipocyte differentiation (*Pparg* deletion) (*Figure 2—figure supplement 6*) (*Berry et al., 2016*; *Jiang et al., 2014*). A potential concern of these methods is that these deletions will occur in other *Acta2+* compartments which could alter beiging potential (*Wendling et al., 2009*). To minimize off-target affects, we conducted these studies in a temporally regulated fashion and proximate to *Adrb3* agonist administration (*Figure 2—figure supplement 6*). Both strategies (*DTA*, *Pparg^fl/fl*) did not appear to alter *Adrb3*-induced beige adipocyte formation or the physiological responses to CL treatment (*Figure 2—figure supplement 6*), indicating that *Acta2+* cells are not a primary source of *Adrb3*-induced beige adipocytes.

Other perivascular sources such as *Myh11+* cells have been shown to fate map into cold-induced beige adipocytes (*Long et al., 2014*). So, we employed the *Myh11^Cre-ERT2* combined with *RosaR26R^RFP* to generate *Myh11^Cre-ERT2*; *R26R^RFP* fate mapping mice. We randomized TM-induced mice to vehicle or CL for seven days and found that CL-induced *Ucp1+* beige adipocytes that were RFP negative (*Figure 2—figure supplement 7*). To further test if *Myh11+* cells could commit to the *Adrb3*-induced beige progenitor lineage, we administered TM to two-month-old *Myh11^Cre-ERT2*; *R26R^RFP* mice and administered vehicle or CL one month post TM for seven days. We found that no new *Adrb3*-induced beige adipocytes emanated from an *Myh11+* smooth muscle cell under this longer time interval (*Figure 2—figure supplement 8*). Collectively, our fate mapping and necessity tests demonstrate that smooth muscle/mural cells, which are required for cold-induced beiging, are not a cellular source of *Adrb3*-induced beige adipocytes.

## *Pdgfra+* cells are a minor subset of progenitors for *Adrb3*-induced beige adipocytes

*Adrb3* agonists can stimulate perivascular *Pdgfra+* fibroblasts within visceral PGW, but not IGW depots, to form beige adipocytes (*Lee et al., 2012*). Yet, *Pdgfra^Cre-ERT2*; *RFP* cells do not appear to generate cold-induced beige adipocytes (*Berry et al., 2016*; *Lee et al., 2015*). As a potential difference between cold- and *Adrb3*-induced beiging, we re-visited if *Pdgfra+* cells are a source of *Adrb3*-induced beige adipocytes by employing the *Pdgfra^Cre-ERT2*; *RFP* model (*Figure 3—figure*

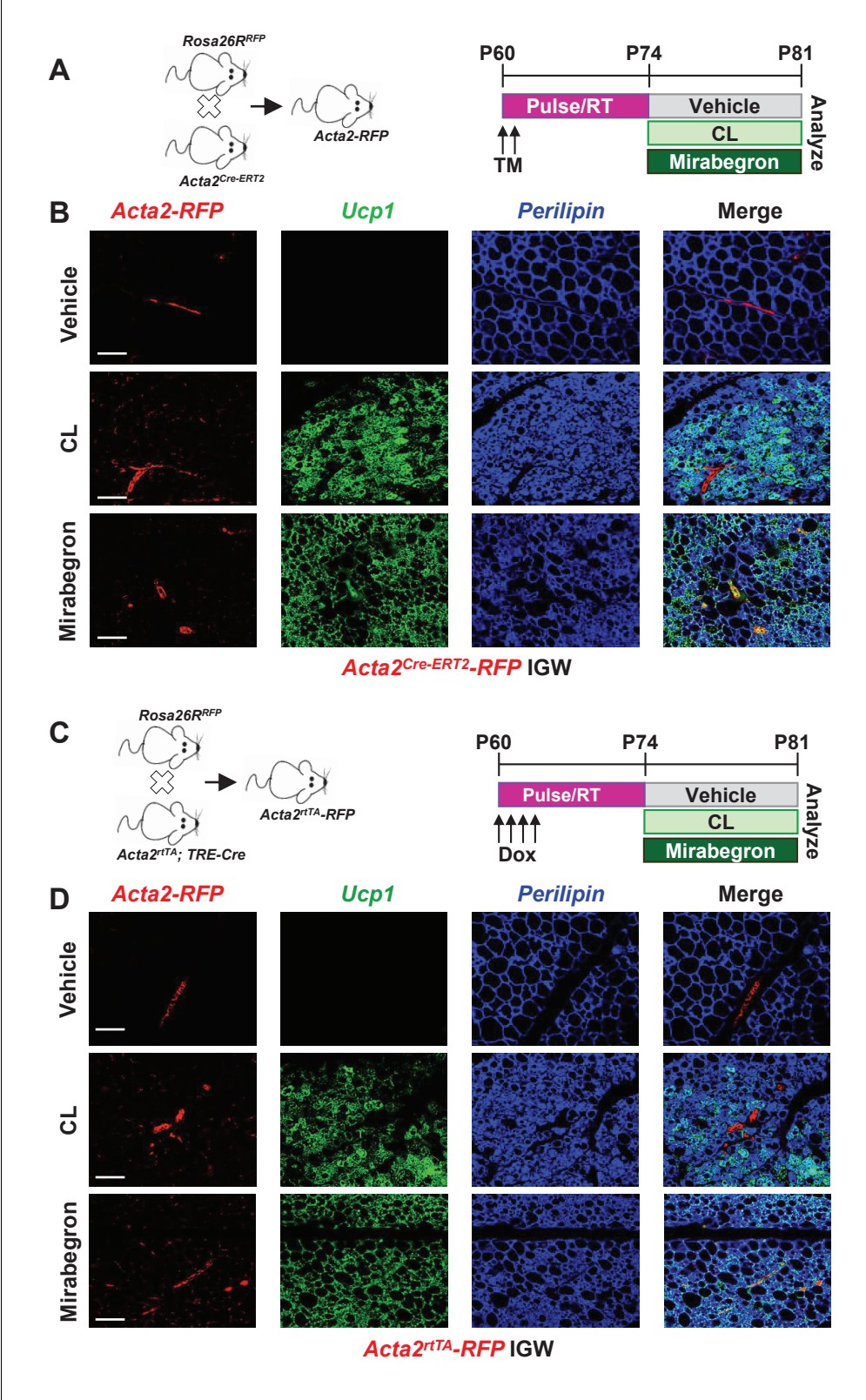

**Figure 2.** Smooth muscle cells are not a source of *Adrb3*-induced beiging. (A) Illustration of mice and experimental regime: two-month-old *Acta2^Cre-ERT2^; RosaR26^RFP^* male mice were administered one dose of TM for two consecutive days. Two weeks post TM, mice were randomized to vehicle, CL or mirabegron for seven days. (B) Representative images of fate mapping analysis of *Acta2^Cre-ERT2^*-RFP+ cells into *Ucp1* beige adipocytes within IGW depots from mice described in (A). (C) Illustration of mice and experimental regime: two-month-old *Acta2^rtTA^; TRE-Cre; RosaR26^RFP^* male mice were

*Figure 2 continued on next page*

*Figure 2 continued*

administered doxycycline (Dox) for four consecutive days. Two weeks post Dox, mice were randomized to vehicle, CL or mirabegron for seven days. (D) Representative images of fate mapping analysis of *Acta2*tTA-RFP+ cells into *Ucp1* beige adipocytes within IGW depots from mice described in (C). Scale bar = 200 μm.

DOI: https://doi.org/10.7554/eLife.30329.005

The following figure supplements are available for figure 2:

**Figure supplement 1.** Induction of beige adipocytes by CL316,243 and mirabegron.

DOI: https://doi.org/10.7554/eLife.30329.006

**Figure supplement 2.** *Acta2+* smooth muscle cells are not a source of *Adrb3* induced beiging.

DOI: https://doi.org/10.7554/eLife.30329.007

**Figure supplement 3.** *Acta2+* smooth muscle cells are a source for cold induced beiging but not for *Adrb3*.

DOI: https://doi.org/10.7554/eLife.30329.008

**Figure supplement 4.** *Acta2+* smooth muscle cells are not a source of *Adrb3* induced beiging at different ages.

DOI: https://doi.org/10.7554/eLife.30329.009

**Figure supplement 5.** *Acta2* genetic tools do not mark or track *Adrb3* induced beiging.

DOI: https://doi.org/10.7554/eLife.30329.010

**Figure supplement 6.** *Acta2 Adrb3* necessity tests.

DOI: https://doi.org/10.7554/eLife.30329.011

**Figure supplement 7.** *Myh11+* smooth muscle cells are not a source of *Adrb3* induced beiging.

DOI: https://doi.org/10.7554/eLife.30329.012

**Figure supplement 8.** *Myh11+* smooth muscle cells are not a source of *Adrb3* induced beiging.

DOI: https://doi.org/10.7554/eLife.30329.013

*supplement 1*). TM induced *Pdgfra*Cre-ERT2; *Rosa26R*RFP two-month-old male mice were randomized to vehicle, CL or MB (*Figure 3—figure supplement 1*). Fate-mapping revealed that roughly 2% of subcutaneous IGW multilocular *Ucp1+* beige adipocytes were RFP positive (*Figure 3A*). Conversely, ~10–15% of perigonadal multilocular *Ucp1+* beige adipocytes were RFP positive (*Figure 3—figure supplement 1*). However, many *Ucp1* antibody positive beige adipocytes were RFP negative, suggesting that *Pdgfra+* cells are not a major source used to generate *Adrb3*-induced beige adipocytes. One potential caveat of inducible systems is recombination efficiency (*Feil et al., 1997*). To examine recombination efficiency, we TM pulsed two-month-old *Pdgfra*Cre-ERT2; *Rosa26R*RFP mice. SV cells were isolated from subcutaneous adipose depots (inguinal and periscapular) and subjected to flow cytometric analysis, 24 hr after the last TM injection. We found that *Pdgfra*-RFP+ cells were 100% positive for *Pdgfra* antibody staining (*Figure 3B*). Conversely, ~70% of *Pdgfra* antibody+ cells were RFP+ indicating a high correspondence between reporter and endogenous *Pdgfra* expression (*Figure 3C,D*). We also performed quantitative real-time PCR analysis of isolated *Pdgfra*-RFP+ cells and found that these cells did not express a smooth muscle signature (*Figure 3E*). Immunohistochemistry also demonstrated that *Pdgfra*-RFP+ cells did not overlap with *Acta2+* cells, as previously reported (*Figure 3—figure supplement 1*) (*Lee et al., 2012*).

To support our *Pdgfra* fate-mapping studies, we incorporated a *Pparg*fl/fl allele with the *Pdgfra*Cre-ERT2 mouse model (denoted *PRa-Pparg*) to block adipogenic action in *Pdgfra+* cells. Two-month-old *PRa-Pparg* male mice were TM-induced two weeks prior to vehicle or CL treatment for seven days (*Figure 3F*). We found that *Adrb3*-induced beige adipocyte formation was unaltered in response to blocking adipogenesis in *Pdgfra+* cells (*Figure 3G–I* and *Figure 3—figure supplement 2*). However, in PGW depots, some beige genes were dampened (*Figure 3J*). Together, it appears, under our conditions, that *Pdgfra+* cells are a minor WAT depot-specific subset of progenitors for *Adrb3*-induced beiging.

## *Pparg*-marked perivascular progenitors do not generate *Adrb3*-induced beige adipocytes

To globally examine if *Adrb3*-induced beige adipocytes were generated from a common white and beige adipose progenitor cell source, we employed the AdipoTrak system (*Pparg*tTA; *TRE-H2B*GFP), which marks the entire adipose lineage (*Jiang et al., 2014*; *Tang et al., 2008*). We Dox suppressed the AdipoTrak system from conception until postnatal day 60 (P60) (*Figure 3—figure supplement 3*). Under these conditions, *Pparg*tTA activity is suppressed and prevents nucleosome incorporation

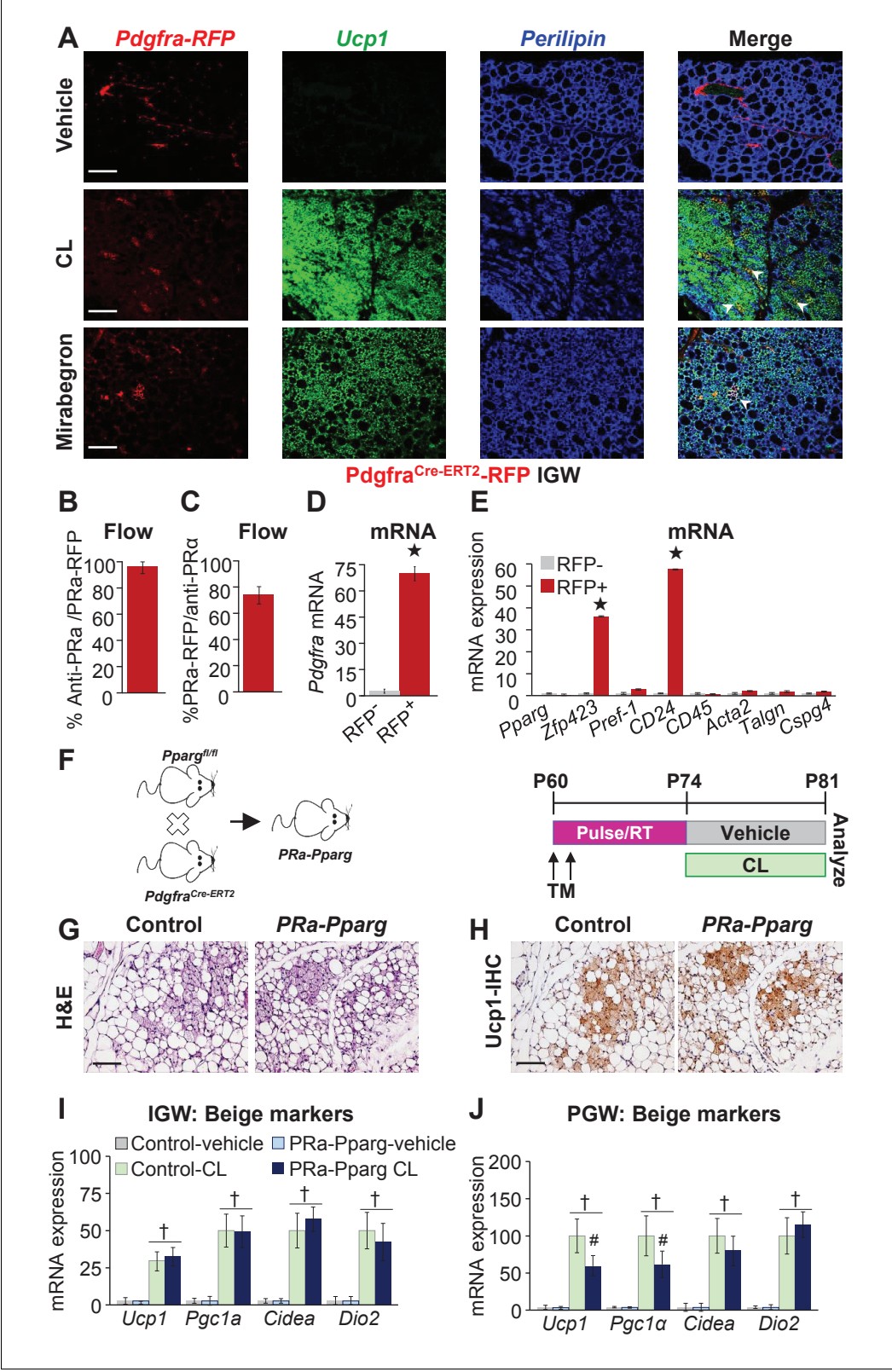

**Figure 3.** *Pdgfra+* cells are not required for *Adrb3*-induced beiging. (**A**) Representative images of *Pdgfra-RFP* fate mapping immunostaining of IGW depots from two-month-old male *Pdgfrα*^Cre-ERT2; *RosaR26*^RFP mice. Mice were administered one dose of tamoxifen (TM) for two consecutive days and after a two-week TM washout period mouse were treated with vehicle, CL or mirabegron for seven days. White arrowheads indicate RFP+ and *Ucp1* +beige adipocytes. (**B**) TM-pulsed *Pdgfra-RFP+* cells were FACS isolated from the WAT SV compartment and antibody stained for *Pdgfra* and

*Figure 3 continued on next page*

Figure 3 continued

analyzed by flow cytometry (n= 6/group). (C) SV cells were isolated from TM-pulsed *Pdgfra-RFP+* mice. Cells were immunostained for *Pdgfra* and examined for *Pdgfra-RFP* positivity by flow cytometry (n= 6/group). (D) Cells described in (B) were FACS isolated and examined for mRNA expression of *Pdgfra*. ★p<0.001 RFP+ compared to RFP- cells (n= 6/group). (E) Cells described in (B) were FACS isolated and mRNA expression of denoted genes were examined. ★p<0.001 RFP+ compared to RFP- cells (n= 6/group). (F) Illustration (left) and experimental procedure (right) used to generate TM-induced two-month-old *Pdgfra*$^{Cre-ERT2}$; *Pparg*$^{fl/fl}$ (*PRa-Pparg*) male mice (n= 8/group). (G) Representative H&E-staining images of IGW depot sections from mice described in (F). (H) Representative *Ucp1* IHC staining of IGW depot sections from mice described in (F). (I) mRNA expression of beige and thermogenic markers from IGW depots from mice described in (F) (n= 6/group in triplicate). (J) mRNA expression of beige and thermogenic markers from PGW depots from mice described in (F). †p<0.05 unpaired t-test, two tailed: *PRa-Pparg* CL treated compared to control CL treated. Scale bar = 200 µm.

DOI: https://doi.org/10.7554/eLife.30329.014

The following figure supplements are available for figure 3:

**Figure supplement 1.** *PDGFRa+* cells are not a major source of *Adrb3* induced beiging.
DOI: https://doi.org/10.7554/eLife.30329.015

**Figure supplement 2.** *PDGFRa+* necessity tests for *Adrb3* induced beiging.
DOI: https://doi.org/10.7554/eLife.30329.016

**Figure supplement 3.** *Adrb3* induced beiging is not generated by an adipose progenitor cell.
DOI: https://doi.org/10.7554/eLife.30329.017

of *H2B*$^{GFP}$. Dox removal, allows *Pparg*$^{tTA}$ to become active and *H2B*$^{GFP}$ is incorporated into proliferating nucleosomes of the progenitor compartment but not into post-mitotic cells such as existing white and beige adipocytes. GFP will be detected in the nuclei of adipocytes if they differentiate from a proliferating GFP-marked progenitor source. Reactivation of the AdipoTrak system after Dox suppression occurs predominantly around the vasculature, and these GFP-labeled progenitors can then be traced into a fraction of cold-induced beige progenitors (*Berry et al., 2016*). However, after CL administration, histological staining revealed that <2% of beige adipocytes were GFP+ (*Figure 3—figure supplement 3*). These data suggest that the proliferating perivascular *Pparg+* adipose progenitors do not generate *Adrb3*-induced beige adipocytes.

## Pre-existing white adipocytes generate many new *Adrb3*-induced beige adipocytes

Transdifferentiation or interconversion of white adipocytes to beige adipocytes has long been considered the standard notion for the beiging phenomena (*Barbatelli et al., 2010*); however, recent studies have challenged this view (*Berry et al., 2016*; *Wang et al., 2013*; *Vishvanath et al., 2016*). Yet, our data thus far have not identified a progenitor cell source for *Adrb3*-induced beiging. Therefore, we further probed if existing white adipocytes could generate *Adrb3*-induced beige adipocytes by generating *Adiponectin*$^{Cre-ERT2}$; *Rosa26R*$^{RFP}$ (*Adpn-RFP*) mice. *Adpn-RFP* marks 100% of pre-existing white and beige adipocytes but does not mark the adipose SV compartment or newly generating adipocytes (*Figure 4A* and *Figure 4—figure supplement 1*) (*Berry et al., 2016*; *Sassmann et al., 2010*). TM-induced *Adpn-RFP* mice were randomized to vehicle, CL, or MB treatment for seven days, two-weeks post TM administration (*Figure 4B*). Beige adipocytes generated by CL or MB were formed from an *Adpn-RFP+* source (*Figure 4C,D*). That is ~75% of all *Adrb3*-induced *Ucp1+* beige adipocytes were RFP+, suggesting that white adipocytes could interconvert to beige adipocytes (*Figure 4E*). These data were confirmed by longer pulse-chase experiments in which we TM induced two-month-old *Adiponectin*$^{Cre-ERT2}$; *Rosa26R*$^{RFP}$ mice and then treated them with CL two months later. Fate mapping assessment demonstrated that ~70% of *Ucp1+* beige adipocytes were RFP+, suggesting that pre-existing white adipocytes are a major source of *Adrb3*-induced beige adipocytes (*Figure 4—figure supplement 1*).

To continue to test the interconversion of white adipocytes to beige adipocytes in response to *Adrb3* agonists, we turned to an in vitro system in which we could test the interconversion of white adipocytes to beige adipocytes. We isolated SV cells from un-induced *Ucp1*$^{Cre-ERT2}$; *RFP* mice (marks only mature beige adipocytes) and were induced with white adipogenic media for seven days. Cell culture-induced white adipocytes were then treated with vehicle, CL, or MB for 4 hr and thermogenic genes were examined (*Figure 4—figure supplement 1*). Vehicle-treated white adipocytes did not express *Ucp1* mRNA nor other thermogenic genes. However, CL- or MB-treated white

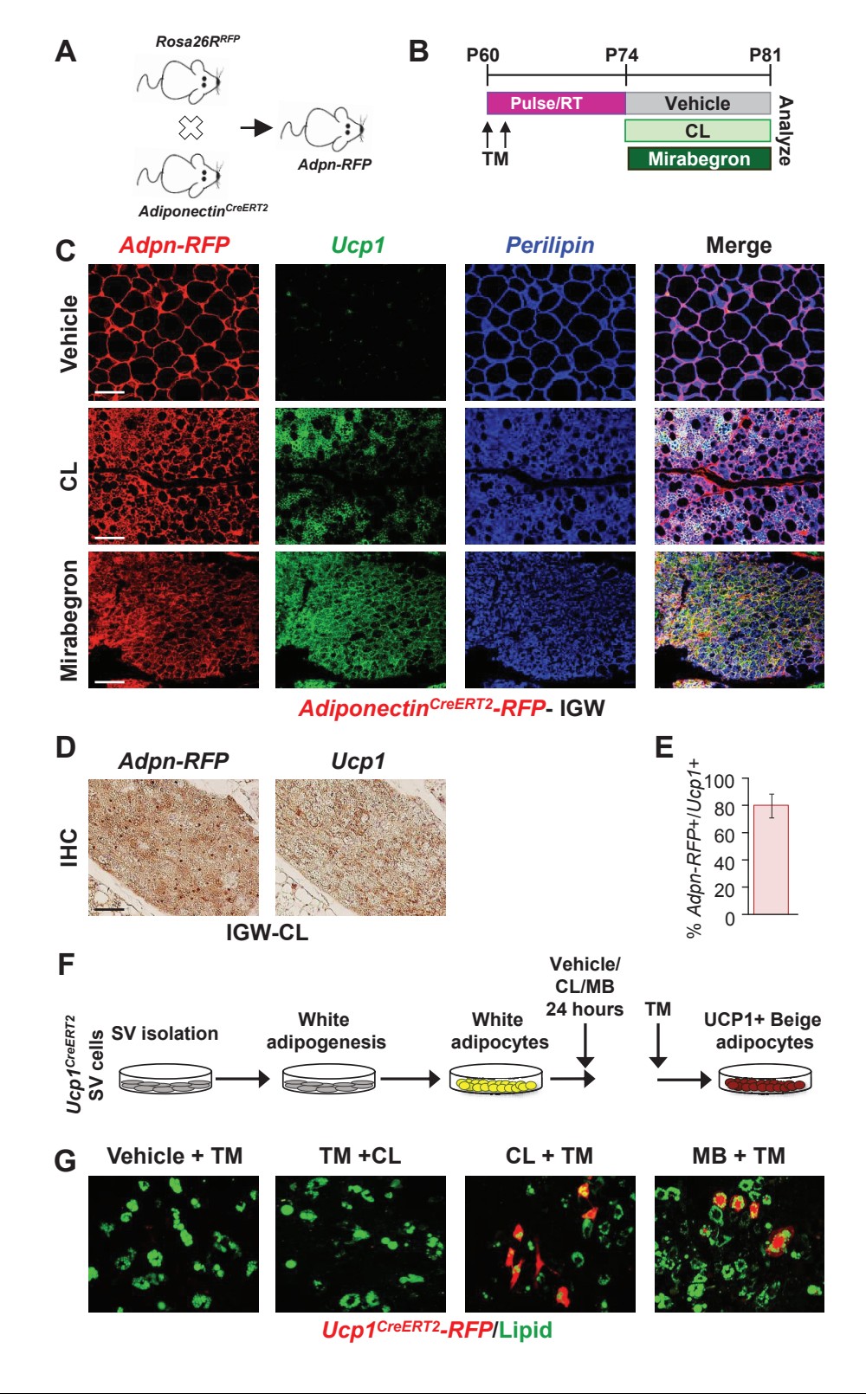

**Figure 4.** Pre-existing white adipocytes are the major source of *Adrb3*-induced beiging. (A) Genetic schema of mice used to generate two-month-old *Adiponectin*^Cre-ERT2^; RFP (*Adpn-RFP*) male mice (*n* = 6/group). (B) Experimental paradigm of TM induction and subsequent treatment with vehicle, CL or mirabegron of *Adpn-RFP* mice. (C) Representative images of *Adpn-RFP* fate mapping and *Ucp1* immunostaining of IGW depot sections from mice described in (B). Scale bar = 200 μm. (D) Representative images of RFP and *Ucp1* IHC of IGW depot sections from mice described in (B). Scale

*Figure 4 continued on next page*

*Figure 4 continued*

bar = 100 µm. (E) Quantitation of co-localized *Adpn-RFP+/Ucp1+* beige adipocytes in response to CL (n= 1000 beige adipocytes). (F) Experimental procedure to stimulate beiging in vitro. (G) Representative images of *Ucp1-RFP* fluorescence co-localization with lipid staining (LipidTox™) after CL treatment of cultured white adipocytes.

DOI: https://doi.org/10.7554/eLife.30329.018

The following figure supplement is available for figure 4:

**Figure supplement 1.** Pre-existing white adipocytes are the major source of *Adrb3* induced beiging.

DOI: https://doi.org/10.7554/eLife.30329.019

adipocytes had elevated expression of *Ucp1* and *Pgc1a*, as previously observed (*Figure 4—figure supplement 1*) (*Cao et al., 2001*). Next, we tested if *Ucp1-RFP* (*Ucp1$^{Cre-ERT2}$; RFP*) expression could be turned on in cultured white adipocytes as a marker of beige adipocyte interconversion. Cultured white adipocytes were treated with three different conditions to monitor *Ucp1-RFP* expression: vehicle and TM, TM and CL, and MB and TM (*Figure 4F*). Under vehicle conditions *Ucp1-RFP* was not expressed. However, treating the cells with CL or MB, and then administering TM, induced the expression of the *Ucp1-RFP* reporter (*Figure 4G*). As a control, we administered TM first then added CL. Under this condition, we did not observe *Ucp1-RFP* reporter expression (*Figure 4G*). Of note, we treated SV cells with CL or MB, and found that *Ucp1* and other thermogenic genes remained undetectable (not shown). Taken together, these data indicate that both pre-existing white adipocytes and cell culture-induced white adipocytes can induce a beige adipocyte morphology and thermogenic program.

## Pre-existing white adipocytes are required for *Adrb3*-induced beige adipocytes

To test if white adipocytes are required to form *Adrb3*-induced beige adipocytes, we incorporated a *Prdm16$^{fl/fl}$* allele with *Adiponectin$^{Cre-ERT2}$; Rosa26R$^{RFP}$* mice (*Figure 5A*). We used *Adiponectin$^{Cre-ERT2}$* to specifically target pre-existing mature white and beige adipocytes to avoid potentially affecting newly evolving progenitors undergoing adipogenesis. Additionally, we are not testing the indisputable role of *Prdm16* in beige adipocyte formation but are testing the requirement of white adipocytes to generate beige adipocytes under *Adrb3* agonists (*Cohen et al., 2014*). In these studies, if white adipocytes are not required then new *Adrb3*-induced beige adipocytes should be observed. We TM induced two-month-old *Adiponectin$^{Cre-ERT2}$; Rosa26R$^{RFP}$; Prdm16$^{fl/fl}$* (*Adpn-Prdm16*) male mice; two weeks later mice were randomized to vehicle, CL, or cold exposed for seven days (*Figure 5B*). We found that cold-exposed Adpn-*Prdm16* mice showed normal beiging (*Figure 5—figure supplement 1*). In contrast, CL-induced beiging was diminished, but not completely absent, in *Adpn-Prdm16* mice (*Figure 5C–G* and *Figure 5—figure supplement 1*). As a next step, we isolated SV cells from un-induced *Adpn-Prdm16* mice. Cells were cultured in white adipogenic media and then administered TM at the end of differentiation to delete *Prdm16* in mature white adipocytes (*Figure 5H*). Adipocytes were treated with vehicle or CL for 24 hr then examined for triglyceride levels and thermogenic gene expression. We found that adipocytes deficient in *Prdm16* were unable to lower triglyceride levels compared to control CL-treated adipocytes (*Figure 5I*). Further, mRNA expression induction of beige and thermogenic genes was dampened in response to *Prdm16* deletion (*Figure 5J*).

To substantiate our white adipocyte necessity results and to test the requirement of *Prdm16* in cold-induced beiging process, we incorporated the *Prdm16$^{fl/fl}$* allele with the *Acta2$^{Cre-ERT2}$* (*Acta2-Prdm16*). We TM induced two-month-old *Acta2*-control and *Acta2-Prdm16* mice; two weeks later mice were cold exposed for one week (*Figure 5—figure supplement 2*). We found that control mice exhibited beiging whereas *Acta2-Prdm16* mice had reduced beiging potential as assessed by rectal temperature, sera glucose, H&E staining, *Ucp1* IHC, *Acta2* fate mapping and beige and thermogenic gene expression (*Figure 5—figure supplement 2*). Collectively, these data suggest that white unilocular adipocytes are a major source and are required for beige adipocyte formation in response to *Adrb3* activation.

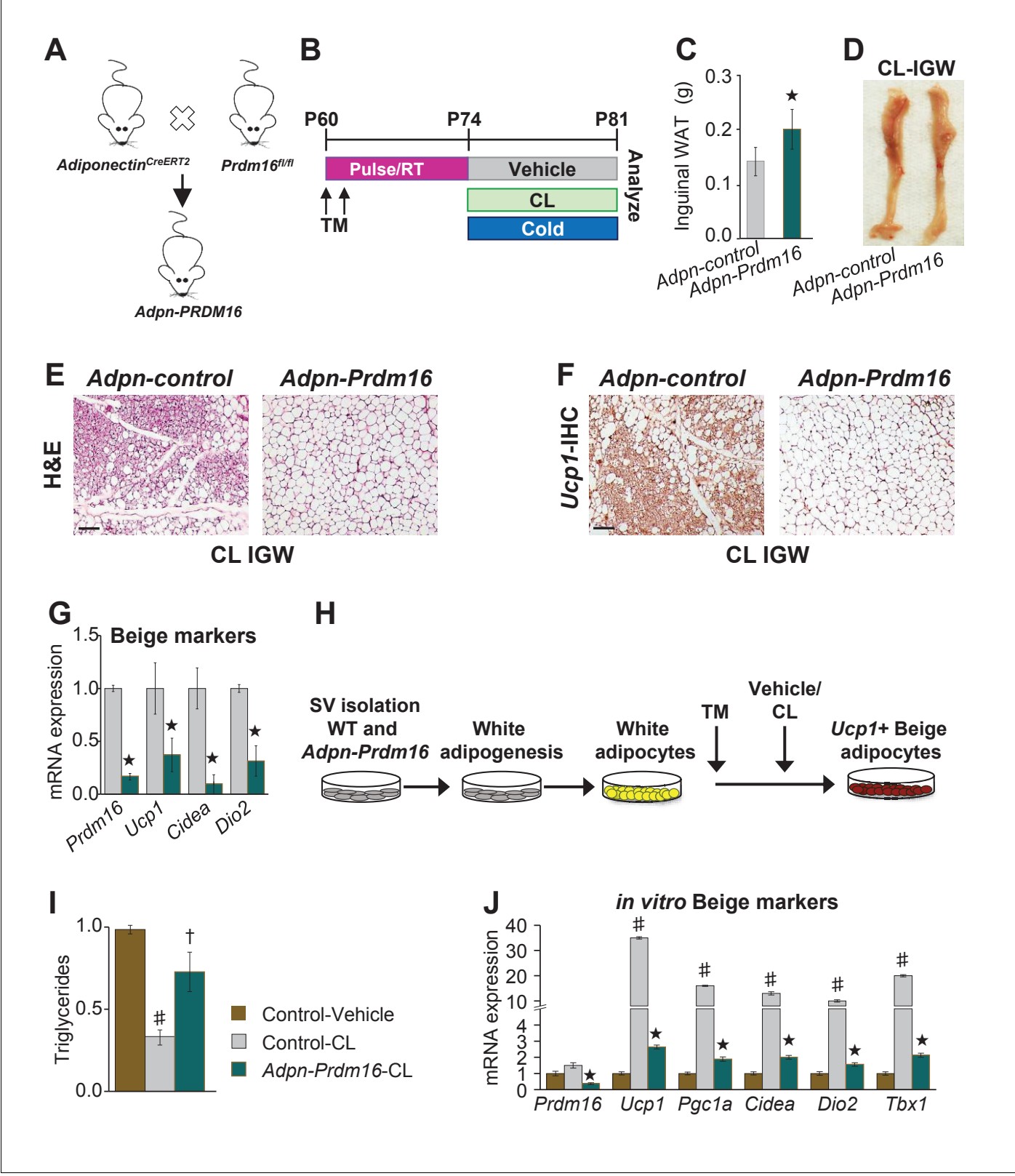

**Figure 5.** Pre-existing white adipocytes are required for *Adrb3*-induced beiging. (A) Illustration of genetic alleles used to generate two-month-old *Adiponectin*$^{Cre-ERT2}$; *RosaR26*$^{RFP}$; *Prdm16*$^{fl/fl}$ (Adiponectin-*Prdm16*) male mice (*n* = 10/group). (B) Experimental procedure to TM induced and treated Adiponectin-*Prdm16* mice. (C) IGW depot weight from mice described in (B). ★p<0.001 unpaired t-test, two tailed: Mutant CL-treated compared to
*Figure 5 continued on next page*

*Figure 5 continued*

control CL-treated mice. (D) Representative photographs of intact IGW depots from mice described in (B). (E) Representative H&E staining of IGW depot sections from mice described in (B). (F) Representative *Ucp1* IHC staining of IGW depot sections from mice described in (B). (G) mRNA expression of beige markers from mice described in (B). ★p<0.001 unpaired t-test, two tailed: Mutant CL- treated compared to control CL-treated mice. (H) Experimental procedure for in vitro beiging. (I) Triglyceride levels in denoted cells after 24 hr vehicle or CL treatment (*n* = 4/group in triplicate). ‡p<0.001 unpaired t-test, two tailed: CL- treated compared to vehicle-treated cells. †p<0.05 unpaired t-test, two tailed: mutant CL-treated compared to vehicle-treated control cells. (J) mRNA expression of beige and thermogenic genes from cells described in (H) (*n* = 6/group in triplicate). ‡p<0.001 unpaired t-test, two tailed: CL-treated compared to vehicle-treated cells. ★p<0.001 unpaired t-test, two tailed: Mutant CL-treated compared to control CL-treated cells. All data are means ±SEM. Scale bar = 200 μm.

DOI: https://doi.org/10.7554/eLife.30329.020

The following figure supplements are available for figure 5:

**Figure supplement 1.** White adipocyte necessity tests for cold and *Adrb3* induced beiging.

DOI: https://doi.org/10.7554/eLife.30329.021

**Figure supplement 2.** *Acta2+* smooth muscle cells are required for cold induced beiging.

DOI: https://doi.org/10.7554/eLife.30329.022

## *Adrb3* does not mediate cold-induced beiging

The above data suggested that *Adrb3* agonists convert white adipocytes to beige adipocytes whereas cold provokes perivascular smooth muscle cells to form beige adipocytes. Previous reports have indicated that *Adrb* signaling mediates beige and brown adipocyte recruitment and thermogenic action in response to cold temperatures and *Adrb3* agonists (*Cannon and Nedergaard, 2004*; *Lidell et al., 2014*). Since cold and *Adrb3* agonists trigger different cell types to form beige adipocytes, we tested if inhibiting *Adrb3* activation altered cold-induced beiging. To test this, *C57BL/6J*-inbred mice were administered vehicle or SR59230A (SR59 1 mg/Kg/day), a *Adrb3* antagonist, for 5 days prior to room temperature, cold exposure or CL administration (*Figure 6A*). SR59 had little to no effect on cold-induced beige adipocyte formation (*Figure 6B–F*). In contrast, blocking *Adrb3* by SR59 reduced the ability of CL to promote beige adipocyte formation (*Figure 6B–G*). Of note, SR59 did not appear to alter brown adipose tissue, morphologically or genetically, in response to either CL or cold exposure (*Figure 6—figure supplement 1*), as observed in *Adrb3* null genetic models (*de Jong et al., 2017*).

## *Adrb*1 mediates cold-induced beiging

In contrast to *Adrb3* expression, which is restricted to mature white and brown adipocytes (*Collins et al., 1994*), other adrenergic receptors such as *Adrb*1 are expressed in the WAT SVF, the source of cold-induced beige progenitors (*Bronnikov et al., 1992*). We too examined if *Adrb*1-3 expression changes in response to white adipogenic media. We found that under white adipogenic conditions, *Adrb*1 and *Adrb*2 were slightly upregulated, and *Adrb3* was significantly upregulated (*Figure 7—figure supplement 1*). As a next step, we performed quantitative real-time PCR analysis on FACS isolated *Acta2-RFP+* cells to examine if *Acta2+* cells expressed any *Adrb*s. We found that *Acta2-RFP+* cells were enriched for *Adrb*1 but not for *Adrb*2, and *Adrb3* was undetectable in both SV compartments (*Figure 7—figure supplement 1*). We also performed quantitative real-time PCR analysis of FACS isolated *PRα-RFP+* fibroblast from WAT depots. We found that neither *Adrb*1 nor *Adrb*2 were enriched in *PRα-RFP+* cells and *Adrb3* was undetectable in both compartments (*Figure 7—figure supplement 1*). Since *Acta2+* cells are enriched in *Adrb*1 this suggested that these cells might be engaged by *Adrb*1 activation to generate cold-induced beige adipocytes. To test this, we inhibited *Adrb*1 by treating mice with vehicle or talinolol (1 mg/Kg/day), a specific *Adrb*1 blocker, for 5 days prior to randomization to room temperature, cold or CL (*Figure 7A*). Under room temperature conditions, talinolol had no effect on temperature, adipose tissue mass, adipose tissue histology and beige adipocyte gene expression (*Figure 7B–G* and *Figure 7—figure supplement 2*). Similarly, talinolol appeared to have no effect on beiging when combined with CL-treatment; that is temperature, adipose tissue mass, histology and beige gene expression were similar between CL alone and CL with talinolol (*Figure 7B–F* and *Figure 7—figure supplement 2*). Conversely, talinolol-treated cold-exposed mice demonstrated an inability to defend body temperature, reduce blood glucose and maintained elevated levels of adipose tissue mass (*Figure 7B,C* and *Figure 7—figure supplement 2*). Further, talinolol-treated cold-exposed mice had reduced beiging

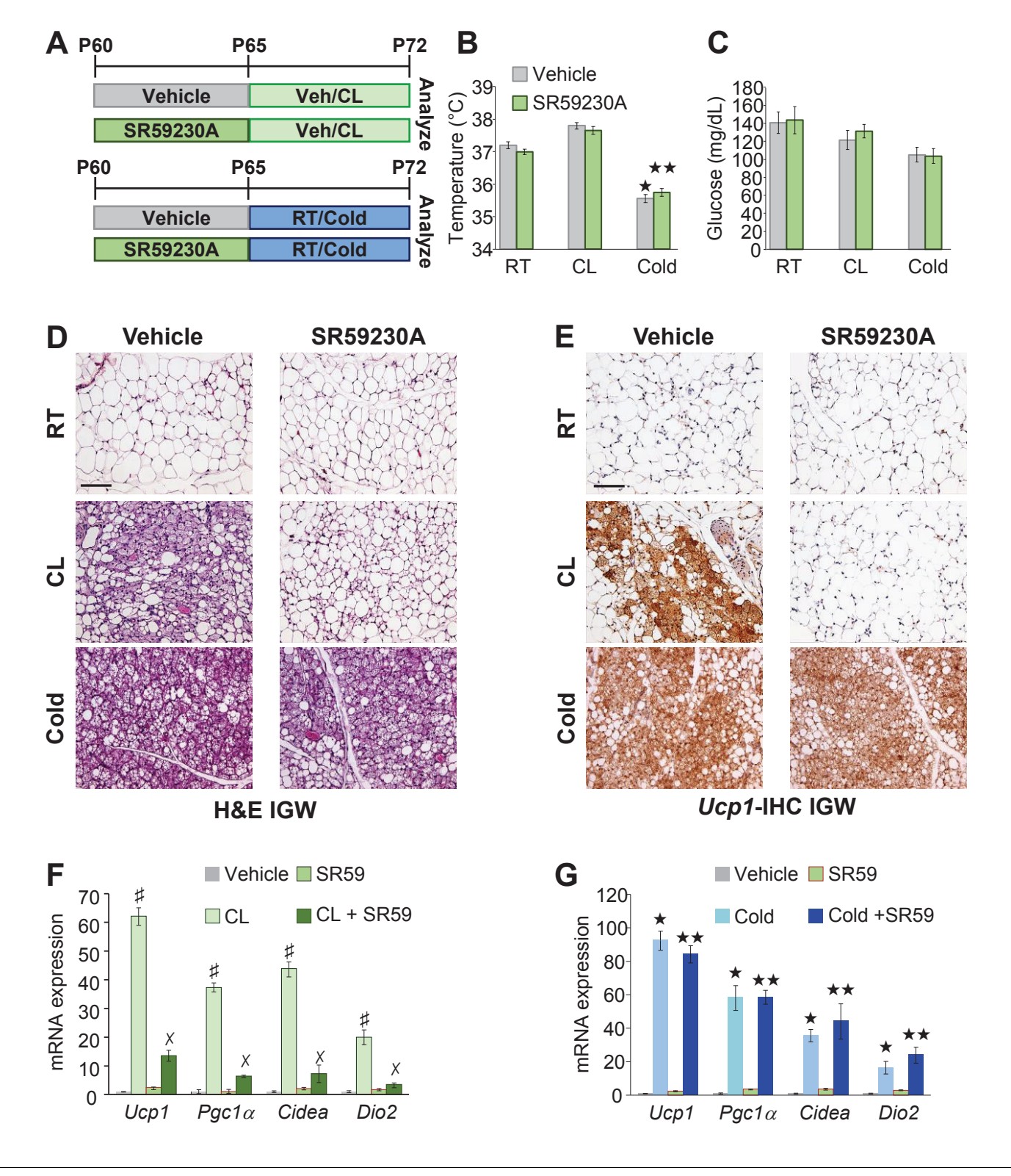

**Figure 6.** *Adrb3* is not required for cold-induced beiging. (A) Illustration of experimental paradigm: Two-month-old male mice were administered vehicle or SR59230A (SR59) for 5 days and then maintained at room temperature, cold exposed or treated with CL for seven days (*n*= 10/group). (B) Rectal temperature from mice described in (A). ★p<0.001 unpaired t-test, two tailed: cold exposed compared to room temperature vehicle-treated mice. ★★p<0.05 unpaired t-test, two tailed: cold + SR59 compared to vehicle room temperature-treated mice. (C) Sera glucose from mice described in

*Figure 6 continued on next page*

*Figure 6 continued*

(A). (D) Representative H&E staining of IGW depot sections from mice described in (A). (E) Representative *Ucp1* IHC staining of IGW depot sections from mice described in (A). (F) mRNA expression of beige and thermogenic genes from mice described in (A) treated with vehicle and SR59 and then maintained at room temperature or cold exposed for seven days (n= 6/group in triplicate). ★p<0.001 unpaired t-test, two tailed: cold exposed compared to room temperature vehicle-treated mice. ★★p<0.05 unpaired t-test, two tailed: cold + SR59 compared to vehicle room temperature-treated mice. (G) mRNA expression of beige and thermogenic genes from mice described in (A) treated with vehicle and SR59 and then administered vehicle or CL for seven days (n= 6/group in triplicate). ‡p<0.01 unpaired t-test, two tailed: CL-treated compared to vehicle-treated mice. ×p<0.01 unpaired t-test, two tailed: CL316 + SR59-treated mice compared to CL-treated mice. All data are means ±SEM. Scale bar = 200 μm.

DOI: https://doi.org/10.7554/eLife.30329.023

The following figure supplement is available for figure 6:

**Figure supplement 1.** Inhibiting *Adrb3* does not alter brown adipose tissue.

DOI: https://doi.org/10.7554/eLife.30329.024

potential as assessed by histology and beige and thermogenic gene expression (*Figure 7D,E and G*). Taken together, these data indicate that *Adrb*1 signaling is involved in cold-induced beige adipocyte formation but not *Adrb3*-induced beiging.

## Discussion

Beige adipocytes have the potential to increase energy expenditure through the consumption of glucose and free fatty acids (*Kajimura et al., 2015*). These potential cellular energy sinks could help address the worldwide epidemic of obesity and its associated metabolic disorders (*Yoneshiro et al., 2013*). Several studies have indicated the presence and induction of metabolically relevant beige adipocytes in humans (*Cypess et al., 2015*; *Saito, 2013*; *Saito et al., 2009*; *van der Lans et al., 2013*; *Wu et al., 2012*; *Yoneshiro et al., 2011b*). For example, administration of mirabegron (MB) increased the resting metabolism of patients and induced the formation of beige adipocytes (*Cypess et al., 2015*). Other studies using cold exposure have seen a similar phenomenon including the induction of beige adipocytes (*Cypess et al., 2009*; *Yoneshiro et al., 2011a*). The full therapeutic potential of beige adipocytes has not been fully realized due, in part, to the lack of genetic tools to mark, track and manipulate the beige adipocyte progenitor cell source; however, the current study appears to overcome some of these barriers.

The cellular sources for cold- and *Adrb3*-induced beige adipocytes have attracted much attention but whether these two beiging stimuli trigger the same or different progenitor sources has remained ambiguous. The same set of tools, used herein, allowed us to identify that *Acta2+* perivascular progenitor cells could serve as an indispensable source of cold-induced beige adipocytes (*Berry et al., 2016*). However, using two independent *Acta2* genetic tools, we were unable to observe *Acta2* beige adipocyte fate mapping in response to *Adrb3* agonists. In addition, we were also unable to observe noticeable fate mapping of *Myh11+* smooth muscle cells, which also serve as a cold-induced beige adipocyte source, into *Adrb3*-induced beige adipocytes. These data suggest that *Adrb3* agonists do not engage smooth muscle-perivascular cells. These findings also correlated with our gene expression data that *Acta2+* cells express *Adrb*1 not *Adrb3*. Moreover, our pharmacological studies indicate that cold-induced beige adipocyte formation relies primarily on *Adrb*1 activation rather than ARDB3 stimulation, which agrees with both our fate mapping and gene expression studies. These findings echo with a recent genetic study that showed that cold-induced beiging is intact in *Adrb3* null mice (*de Jong et al., 2017*). However, other studies have demonstrated that *Adrb3* null mice have less beiging potential (*Barbatelli et al., 2010*; *Jimenez et al., 2003*). The underlying differences between these studies may be, in part, due to genetic backgrounds of the null mice, further studies would be required.

Our studies support the notion that *Adrb3*-induced beige adipocytes originate from white adipocytes. Using *Adiponectin*^Cre-ERT2^; *RFP* fate mapping studies, we found that the majority of *Adrb3*-induced beige adipocytes emanate from these pre-existing adipocytes. However, cold relies on *Acta2+* mural cells. WAT necessity tests using *Adiponectin*^Cre-ERT2^; *Prdm16* demonstrated that pre-existing white adipocytes are required for *Adrb3*-induced beige adipocyte formation. A previous study demonstrated that deleting *Prdm16* within the WAT lineage sufficiently blocked beige adipocyte formation from both cold exposure and *Adrb3* agonists (*Cohen et al., 2014*). The major

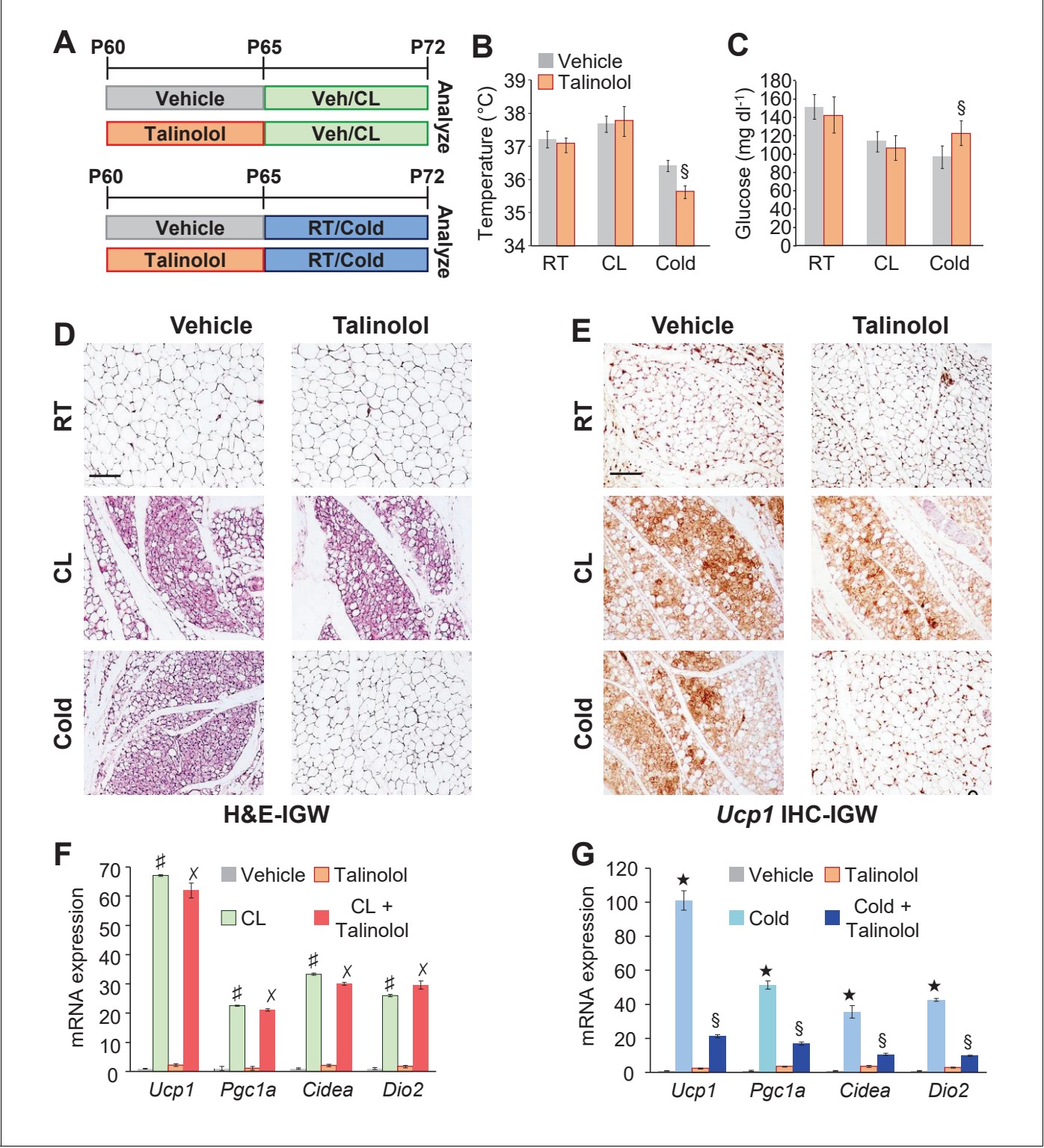

**Figure 7.** *Adrb*1 is required for cold-induced beiging. (**A**) Experimental paradigm: Two-month-old male mice were first treated with vehicle or talinolol (1 mg/Kg/day) for five days. Mice were then maintained at room temperature, treated with CL or cold exposed for seven days (*n*= 10/group). (**B**) Rectal temperature from mice described in (**A**). §p<0.05 unpaired t-test, two tailed: cold + talinolol compared to cold + vehicle. (**C**) Sera glucose from mice described in (**A**). §p<0.05 unpaired t-test, two tailed: cold + talinolol compared to cold + vehicle. (**D**) Representative H&E staining of IGW depot sections from mice described in (**A**). (**E**) Representative *Ucp1* IHC staining of IGW depot sections from mice described in (**A**). (**F**) mRNA expression of beige and thermogenic genes from mice described in (**A**) treated with vehicle and talinolol and then administered vehicle or CL for seven days. ♯

*Figure 7 continued on next page*

*Figure 7 continued*

p<0.001 unpaired t-test, two tailed: CL-treated mice compared to vehicle-treated mice. ×p<0.001 unpaired t-test, two tailed: CL + Talinolol compared to vehicle-treated mice. (G) mRNA expression of beige and thermogenic genes from mice described in (A) treated with vehicle and talinolol and then maintained at room temperature or cold exposed for seven days. ★p<0.001 unpaired t-test, two tailed: cold compared to room temperature vehicle-treated mice. §p<0.05 unpaired t-test, two tailed: cold + talinolol compared to cold + vehicle. All data are means ±SEM. Scale bar = 200 μm.
DOI: https://doi.org/10.7554/eLife.30329.025
The following figure supplements are available for figure 7:

**Figure supplement 1.** *Adrb* expression.
DOI: https://doi.org/10.7554/eLife.30329.026
**Figure supplement 2.** *Adrb1* is required for cold-induced beiging.
DOI: https://doi.org/10.7554/eLife.30329.027

difference between the two studies is the use of the temporal *Adiponectin*$^{Cre-ERT2}$ genetic tool compared to the global *Adiponectin*$^{Cre}$ genetic tool. In our studies using the *Adiponectin*$^{Cre-ERT2}$, *Prdm16* is specifically deleted in all pre-existing adipocytes (beige and white) and not in newly generated adipocytes. In contrast, using *Adiponectin*$^{Cre}$ model, *Prdm16* is continuously deleted in all pre-existing and newly developing adipocytes (*Jeffery et al., 2014*; *Shao et al., 2016*; *Wang et al., 2013*), meaning that under all conditions, cold or *Adrb3* agonist, beige adipogenesis is disrupted. Yet, fate mapping limitations aside, the *Adiponectin*$^{Cre}$ tool identified the critical role of *Prdm16* in beige adipogenesis.

Classical biochemical and pharmacokinetic studies have indicated that *Adrb1* activation is not coupled to thermogenesis in brown adipocytes. In agreement, our data imply that *Adrb1* is critical for *Acta2+* perivascular cells to leave their vascular niche and form beige adipocytes. However, using pharmacological agents such as talinolol, could have systemic affects that may alter beiging potential independent of *Adrb*1 blockade within WAT. Pharmacokinetically, talinolol is highly hydrophilic with a half-life of 6–7 hr and has difficulty crossing the blood-brain barrier (*Neil-Dwyer et al., 1981*; *Sourgens et al., 2003*). Of note, mice were exposed to the cold or CL, 24 hr post talinolol treatment. Further investigation into the downstream signaling of *Adrb*1 in *Acta2+* cells in response to the cold could provide important insight into potential mechanisms and therapeutic targets to stimulate cold-induced beiging perhaps in the absence of cold treatment.

The current study demonstrates that several cell types can form multilocular beige-like adipocytes within white adipose depots. This is consistent with other reports that *Pdgfra+* perivascular cells as well other non-adipocyte sources generate beige adipocytes (*Wang et al., 2013*; *Lee et al., 2012*). Additionally, other studies have suggested that beige and white adipocytes are bi-potential; that is beige adipocytes become white adipocytes but can revert to beige adipocytes in response to a beiging stimulus (*Rosenwald et al., 2013*; *Altshuler-Keylin et al., 2016*). Our studies resonate with these previous findings; however, the data herein suggest that a majority of *Adrb3*-induced beige adipocytes are generated from mature white adipocytes. A potential limitation of these study is that we cannot decipher if these white adipocytes that converted to *Adrb3*-induced beige adipocytes were once beige. Further studies into genetic program and memory could help elucidate how these cells interconvert in response to sympathetic activation. These studies also raise another interesting question; do beige adipocytes that form in response to cold exposure or by *Adrb3* agonists activate thermogenesis in an analogous manner? That is; does cold temperatures invoke the same thermogenic program as *Adrb3* agonist? More studies directed at examining beige metabolic properties could provide pivotal insight into which beige stimuli would be more clinically germane regarding energy utilization. Also do these differently induced beige adipocytes have different genetic signatures? Towards this end, a recent study showed that beige adipocytes generated by CL compared to beige adipocytes produced by roscovitine/rosiglitazone had unique genetic signatures suggesting that not all beige adipocytes are created equally (*Wang et al., 2016*). Hence, further research into the genetic regulation and programs could yield answers to these questions.

In summary, using several genetic models encompassing several proposed sources of *Adrb3*-induced beige adipocytes, we found that white adipocytes generate most *Adrb3*-induced beige adipocytes. Further, we found that cold-induced beiging relies on *Adrb1* activation, and not *Adrb3*, to engage and license smooth muscle cells to form beige adipocytes. These data highlight that several

cellular sources exist for two different beiging stimuli and that not all beiging stimuli should be considered equal.

## Materials and methods

### Animals

All animals were maintained under the ethical guidelines of the UT Southwestern Medical Center Animal Care and Use Committee according to NIH guidelines under the protocol number 2016–101336. Mice were housed in a 12:12 light:dark cycle and chow and water were provided *ad libitum*. Experiments were performed on two or six-month-old male mice unless otherwise stated. For all experiments, mice were randomized to their respective groups without restrictions. All mouse lines were backcrossed on a *C57BL/6J- 129S1/SvlmJ* mixed background for at least nine generations. AdipoTrak mice were previously established in our lab (*Tang et al., 2008*). *C57BL/6J* (stock no: 000664), *Adiponectin$^{Cre-ERT2}$* (stock no: 025124), *Myh11$^{Cre-ERT2}$* (stock no: 019079), *Pparg$^{fl/fl}$* (stock no: 004584), *Prdm16$^{fl/fl}$* (stock no: 024992), *Rosa26R$^{DTA}$* (stock no: 006331), and *Rosa26R$^{RFP}$* (stock no: 007908) mice were obtained from the Jackson Laboratory. *Acta2$^{Cre-ERT2}$* mice were generously provided by Dr. Pierre Chambon. *Acta2$^{rtTA}$* mice were generously provided by Dr. Beverly Rothermel. Drs. Sean Morrison and Bill Richardson generously provided the *Pdgfra$^{Cre-ERT2}$* mice. *Ucp1$^{Cre-ERT2}$* were generously provided by Dr. Eric Olson. Cre recombination was induced by administering one dose of tamoxifen dissolved in sunflower oil (Sigma, 50 mg/Kg interperitoneally injection) for two consecutive days. rtTA activation was induced by Doxycycline (0.5 mg/ml in 1% sucrose) provided in the drinking water and protected from light, and it was changed every 2–3 days. For cold experiments, mice were placed in a 6.5°C cold chamber or maintained at room temperature (23–25°C) for seven days.

### Pharmacological administration

CL316,243 was purchased from Tocris and dissolved in water. CL316,243 was administered at one dose (1 mg/Kg/day) for seven consecutive days by interperitoneal injections (IP). Mirabegron was purchased from Cayman Chemical and dissolved in water and was administered at one dose (1 mg/Kg/day) for seven consecutive days by IP. Talinolol was purchased from Cayman Chemical and was dissolved in DMSO. Administration solution was dissolved further in water (DMSO ~5%) and was administered at one dose (1 mg/Kg/day) for five consecutive days by IP. SR59230A was purchased from Sigma-Aldrich and was dissolved in DMSO. Administration solution was dissolved further in water (DMSO ~5%) and was administered at one dose (1 mg/Kg/day) for five consecutive days by IP.

### Stromal vascular fractionation isolation and cell culture

No cell lines were used. Stromal-vascular (SV) cells were isolated from pooled subcutaneous (inguinal, interscapular) white adipose tissues for fractionation, unless indicated otherwise. After 2 hr of slow shaking in isolation buffer (100 mM HEPES pH7.4, 120 mM NaCl, 50 mM KCl, 5 mM glucose, 1 mM CaCl2, 1.5% BSA) containing 1 mg/ml collagenase at 37°C. The suspension was then spun at 800 g for 10 min and the pellet contained a crude SV fraction. The SV pellet was then re-suspended in erythrocyte lysis buffer (0.83% NH4Cl in H$_2$O) for 5 min, spun at 800xg for 5 min. The pellet was washed once in 1X PBS, re-suspended and passed through 40 µm mesh. Isolated SV cells were cultured in DMEM supplemented with 10% FBS with 1% penicillin and streptomycin. White adipogenesis was induced by treating confluent cells with DMEM containing 10% FBS, insulin (0.5 µg/ml), dexamethasone (5 µM), and isobutylmethylxanthine (0.5 mM). To induce beige and thermogenic genes, cells were treated with 5 µM CL316,243 or 5 µM mirabegron for 4 or 24 hr. mRNA was harvested or cells were imaged (*Wu et al., 2012*). Triglyceride accumulation was performed using a kit from ZenBio (*Berry and Noy, 2009*; *Berry et al., 2010*) and manufacturer's protocol was followed.

### Flow cytometry

SV cells were isolated as above and washed, centrifuged at 1200 g for 5 min, and analyzed with a FACScans analyzer or sorted with a BD FACS Aria operated by the UT Southwestern Flow Cytometry Core. Data analysis was performed using BD FACS Diva and FlowJo software. For RFP+ sorting, live SV cells from *Acta2$^{Cre-ERT2}$; R26R$^{RFP}$*, or *Pdgfra$^{Cre-ERT2}$; R26R$^{RFP}$*, or *Adiponectin$^{Cre-ERT2}$; R26R$^{RFP}$*

mice were sorted based on native fluorescence (RFP). The SV cells from TM-induced *Acta2*<sup>Cre-ERT2</sup> or *Pdgfra*<sup>Cre-ERT2</sup> (without RFP) control mice were used to determine background fluorescence levels. SV cells were incubated with primary antibodies on ice for 30 min and then washed twice with the staining buffer and incubated with secondary antibody for another 30 min on ice before flow cytometry analysis. Primary antibodies include: rabbit-anti-*Pdgfrα* (1:100, Santa Cruz Biotechnology).

## Quantitative real-time PCR (qPCR)

Total RNA was extracted using TRIzol (Invitrogen: item no: 15596026) from either mouse tissues or cells. Mouse tissues (*n*= 6 individual tissues) and cells (*n*= 4 individual wells) were pooled and analyzed in technical quadruplicates. These experiments were performed on three independent cohorts. cDNA synthesis was performed using RNA to cDNA high capacity kit (Invitrogen: item no: 4387406). Gene expression was analyzed using Power SYBR Green PCR Master Mix with ABI 7500 Real-Time PCR System. qPCR values were normalized by 18 s rRNA expression. Primer sequences are available in **Supplementary file 1**.

## Histological staining

Adipose tissues were fixed in 10% formalin, dehydrated, embedded in paraffin, and sectioned with a Microm HM 325 microtome at 5–15 μm thickness. For immunofluorescence staining, paraffin sections were preincubated with permeabilization buffer (0.3% Triton X-100 in PBS) for 30 min at room temperature and then incubated sequentially with primary antibody (4°C, overnight) and secondary antibody (2 hr at room temperature), all in blocking buffer (5% normal donkey serum in 1X PBS). Antibodies used for immunostaining are: rabbit-anti-*Ucp1* (1:500, Abcam), mouse-anti-RFP (1:200, Clontech), mouse-anti-*Acta2* (1:500, Abcam), rabbit-anti-Pdgfrα (1:100, Santa Cruz Biotechnology), and goat-anti-*Perilipin* (1:500, Abcam). Secondary antibodies, Alexa Fluor 488 donkey anti-rabbit, cy3 donkey anti-mouse, and Alexa Fluor 647 donkey anti-goat, were from Jackson ImmunoResearch. All secondary antibodies were used at a 1:500 dilution. For immunohistochemistry staining, slides were deparaffinized and rehydrated before heat-induced antigen retrieval. Antigens were detected using primary antibody and in conjunction with an HRP/DAB (ABC) detection kit (Abcam; ab64264) according to the manufacturer's instructions (R&D Systems). Immunostained images were collected on a Zeiss LSM500 confocal microscope, or a Lecia DMi6. For quantification of images, two independent observers assessed three random fields in 10 random sections from at least three mice per cohort and used Image J to quantify co-localization.

## Metabolic phenotyping experiments

Temperature was monitored daily using a rectal probe (Physitemp). The probe was lubricated with glycerol and was inserted 1.27 centimeters (1/2 inch) and temperature was measured when stabilized. Glucose monitoring: tail blood glucose levels were measured immediately after cold exposure (9am CST) with a Contour glucometer (Bayer).

## Statistical analysis

Statistical significance was assessed by an unpaired Student's *t*-test using Origin Labs 8.1 software, Excel or GraphPad Prism 6. $p < 0.05$ was considered statistically significant. Mouse experiments were performed in biological triplicate with at least six mice/group and results are expressed as means ± SEM. No power analysis was used to calculate samples size rather sample size was calculated on historical fate-mapping and metabolic studies from the ours and other's research group. For fate-mapping experiments, a minimum of 6 mice/group were analyzed replicated thrice. For metabolic studies, a minimum of 8 mice/group were analyzed replicated thrice. Experiments were performed on male mice at denoted ages. No mice were excluded from the study unless visible fight wounds were observed. Cell culture experiments were collected from three or four independent cultures for each sample and pooled.

## Additional information

### Competing interests

Jonathan M Graff: Co-founder and shareholder of Reata Pharmaceuticals. Reata Pharmaceuticals has no financial interest in this study. The other authors declare that no competing interests exist.

### Funding

| Funder | Grant reference number | Author |
| --- | --- | --- |
| National Institute of Diabetes and Digestive and Kidney Diseases | K01 DK109027 | Daniel C Berry |
| National Institute of Diabetes and Digestive and Kidney Diseases | K01 DK111771 | Yuwei Jiang |
| National Institute of Diabetes and Digestive and Kidney Diseases | R01 DK088220 | John M Graff |
| National Institute of Diabetes and Digestive and Kidney Diseases | R01 DK064261 | John M Graff |
| National Institute of Diabetes and Digestive and Kidney Diseases | R01 DK066556 | John M Graff |

The funders had no role in study design, data collection and interpretation, or the decision to submit the work for publication.

### Author contributions

Yuwei Jiang, Conceptualization, Data curation, Formal analysis, Investigation, Methodology, Writing—original draft, Writing—review and editing; Daniel C Berry, Conceptualization, Data curation, Formal analysis, Supervision, Investigation, Methodology, Writing—original draft, Writing—review and editing; Jonathan M Graff, Supervision

### Author ORCIDs

Yuwei Jiang http://orcid.org/0000-0002-2132-0279
Daniel C Berry http://orcid.org/0000-0002-5200-1182

### Ethics

Animal experimentation: This study was performed in accordance with the recommendations in the Guide for the Care and Use of Laboratory Animals of the National Institutes of Health. All animals were maintained under the approved protocols and ethical guidelines of the UT Southwestern Medical Center Animal Care and Use Committee under the protocol number 2016–101336.

### Decision letter and Author response

Decision letter https://doi.org/10.7554/eLife.30329.030
Author response https://doi.org/10.7554/eLife.30329.031

## Additional files

### Supplementary files

• Supplementary file 1. qPCR primer sequences utilized for gene expression analysis.
DOI: https://doi.org/10.7554/eLife.30329.028

• Transparent reporting form
DOI: https://doi.org/10.7554/eLife.30329.029

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
