## [Decision Letter]

Thank you for submitting your article "Distinct cellular and molecular mechanisms for cold and β3 adrenergic receptor induced beige adipocyte formation" for consideration by *eLife*. Your article has been favorably evaluated by Mark McCarthy (Senior Editor) and three reviewers, one of whom is a member of our Board of Reviewing Editors. The reviewers have opted to remain anonymous.

The reviewers have discussed the reviews with one another and the Reviewing Editor has drafted this decision to help you prepare a revised submission.

Summary:

Berry and colleagues present interesting findings about the ontogeny of beige fat cells. These cells are inducible, and most studies have used cold as a stimulus to promote beige fat biogenesis. However, other stimuli are also capable of inducing beige fat, such as β3 adrenergic agonists. Here the authors show that while cold-induced beige adipocytes come from perivascular SMA+ cells, β3 agonists promote beige fat through a different mechanism: conversion from mature white adipocytes. Moreover, the authors show that cold seems to induce beige fat via β1 adrenergic receptor (ADRB1) activation, another distinction between these two stimuli.

The reviewers were in agreement that the work was potentially of interest to the general audience of *eLife*. The studies were judged to be technically sound and the reviewers believed that the majority of the conclusions drawn were supported by the data. At the same time, the review process identified several opportunities to strengthen the work.

Essential revisions:

1) For the fate mapping experiments in Figure 2, Figure 3, Figure 4, and Figure 5 the authors use an approach in which animals are pulse labeled with tamoxifen at p 60 and then a treatment is administered 14 days later with subsequent analysis. Since we do not know the rate of development/turnover of beige adipocytes, how can the authors be sure this is the appropriate time point to sample? For example, in Figure 2, they conclude that β3 agonists do not promote beige fat via the development of SMA+ cells. However, isn't it possible that with a longer time interval, they might actually see a number of SMA+/UCP1+ cells? If data in any of these experiments is different at longer time points, it could change the conclusions of the paper dramatically. For that reason, it would be more convincing if at least some of the experiments were also done with a longer time interval between pulsing and analysis.

2) Related to the above point, a longer-term cold exposure experiment is particularly important. Cold exposure causes release of norepinephrine from sympathetic nerves, which will act on both ADRB1 and ADRB3. If the current manuscript is correct, in cold exposure, beiging should have a rapid phase of ADRB3-mediated white-to-beige conversion, followed by a slower de novo ADRB1-mediated beige adipocyte differentiation phase. This is not clear from their data because they only look at the effect of cold at 1 week. They should do a longer time course experiment to see if cold induces a biphasic response, as would be predicted.

3) The images shown throughout the paper are relatively high power, focusing on a limited portion of the tissue. It would be helpful, at least in some cases, if lower power images could also be shown to ensure that these conclusions are not biased by just focusing on the region that best proves a certain point.

4) The authors state: "cold exposed Adpn-PRDM16 mice showed normal beiging, as expected from previous studies (Figure 5—figure supplement 1) (Berry et al., 2016)". However, there are no data from PRDM16 KO mice in Berry et al. (2016). In addition, Cohen et al. (2014) showed that Adpn-PRDM16 mice are defective in both cold- and CL^-^induced browning in iWAT. The authors need to clarify how their work fits into the scheme of what's been published previously.

5) Conflicting data have been reported for beiging in Adrb3 KO mice (Jimenez, Eur. J Biochem. 2003, Barbatelli, AJPEM, 2010, de Jong, AJPEM, 2017). The authors should address how their data contribute to the previous controversy.

6) Blocking ADRB1 by IP injection leads to systemic effects, especially profound effects on cardiac function. Could any of the observations on beiging be secondary to other physiological changes?

7) The title of the manuscript covers both cold- and ADRB3-induced beige cell formation, however the majority of the studies focused on identifying the origin of ADRB3-induced beige adipocytes. There is minimal data presented on the progenitor of cold-induced beige adipocytes. The title could be modified and narrowed to better match the scope of this study.

---

## [Author Response]

Essential revisions:1) For the fate mapping experiments in Figure 2, Figure 3, Figure 4, and Figure 5, the authors use an approach in which animals are pulse labeled with tamoxifen at p 60 and then a treatment is administered 14 days later with subsequent analysis. Since we do not know the rate of development/turnover of beige adipocytes, how can the authors be sure this is the appropriate time point to sample? For example, in Figure 2, they conclude that β3 agonists do not promote beige fat via the development of SMA+ cells. However, isn't it possible that with a longer time interval, they might actually see a number of SMA+/UCP1+ cells? If data in any of these experiments is different at longer time points, it could change the conclusions of the paper dramatically. For that reason, it would be more convincing if at least some of the experiments were also done with a longer time interval between pulsing and analysis.

We thank the reviewers for noting this concern and limitation of our genetic model. Our previous work identified that SMA marks both the white and beige lineages or a common precursor. Fate mapping studies from tamoxifen induced SMA-CreERT2; RFP mice, showed that within 30 days, post labeling of SMA+ mural cells, white adipocytes were generated from this source. The reviewers adequately questioned whether waiting longer after tamoxifen injection might allow SMA+ perivascular cells to commit to an ADRB3 induced beige progenitor. However, given the information above, if tamoxifen were administered and then several weeks or months passed then new SMA-labeled white adipocytes would be generated under room temperature (23-25C) conditions throughout the fat tissue. Subsequently, if mice were then subjected to CL316,243 after this longer duration then new beige adipocytes could emanate from an SMA-labeled white adipocyte or from an SMA+ mural cells. Therefore, the conclusions generated from these experiments could not solely be based on progenitor or adipocyte sources but rather mixed and uninterpretable. We apologize that we could not perform these experiments due to the ambiguous nature of the genetic tools and because we do agree with the reviewers that is an important notion.

However, we did try to address the reviewer’s comment in another manner which was to label SMA cells at various ages (postnatal day-P30, P60, P90 and P180) and then administer CL316,243 14 days later. Again, we found that under our conditions that no beige adipocytes were generated from an SMA cell source (Figure 2—figure supplement 4). Since both SMA and Myh11 cells are used in cold induced beiging, we tested a longer time interval in Myh11; RFP fate mapping mice. We administered TM at P60 and waited 30 days prior to administration of vehicle or CL for seven days. Again, we did not observe fate mapping into ADRB3-induced beige adipocytes (Figure 2—figure supplement 8). As an extension of this and to test if a progenitor cell was required during these longer time intervals, we labeled, by tamoxifen, all adipocytes using Adiponectin-CreERT2; RFP and then administered CL316,243 two months later. We found that ~70% of all UCP1+ beige adipocytes generated by CL316,243 emanated from an adipocytes source. Together, this new dataset bolsters the notion that mature white adipocytes are the major contributor to the ADRB3 beiging process.

2) Related to the above point, a longer-term cold exposure experiment is particularly important. Cold exposure causes release of norepinephrine from sympathetic nerves, which will act on both ADRB1 and ADRB3. If the current manuscript is correct, in cold exposure, beiging should have a rapid phase of ADRB3-mediated white-to-beige conversion, followed by a slower de novo ADRB1-mediated beige adipocyte differentiation phase. This is not clear from their data because they only look at the effect of cold at 1 week. They should do a longer time course experiment to see if cold induces a biphasic response, as would be predicted.

We thank the reviewers for these suggestions. First, as recommended by the reviewers, to continue to test the requirement of mural cells to cold and ADRB3 induced beiging, we performed a 2-week cold exposure study and found that 60% of all UCP1+ beige adipocytes were from an SMA-RFP+ mural cell origin which is comparable to 1 week of cold exposure. Contemporaneously, we also treated TM-induced SMA-CreERT2; RFP mice for 2 weeks with CL316,243 to test if chronic exposure to ADRB3 agonist could stimulate these cells to leave their vascular niche and form beige adipocytes. We found that SMA-RFP+ cells did not form more than 5% of beige adipocytes throughout the adipose depots. Overall, the data appear to show that SMA+ mural cells are not a source for ADRB3 induced beige adipogenesis but function in response to cold.

We thank the reviewers for projecting a biphasic model of the beiging process. The reviewers noted that ADRB activation might be biphasic in that ADRB3 is activated first stimulating a white-to-beige adipocyte interconversion and then secondarily ADRB1 would be activated and simulate the progenitor cells. However, our ADRB3 pharmacological inhibition, as well as genetic studies (de Jong, AJPEM, 2017), demonstrate that ADRB3 has a minor role, overall, in cold induced beiging, irrespective of timing. Rather these data support the notion that ADRB3 activation, independent of cold exposure, mediates the conversion of white-to-beige adipocytes and suggest a different means to induce beige adipocytes formation.

3) The images shown throughout the paper are relatively high power, focusing on a limited portion of the tissue. It would be helpful, at least in some cases, if lower power images could also be shown to ensure that these conclusions are not biased by just focusing on the region that best proves a certain point.

We have added several lower magnification images throughout the manuscript to document cold-induced and ADRB3 induced beige adipogenesis (Figure 1—figure supplement 1; Figure 2—figure supplement 2; Figure 2—figure supplement 5; Figure 3—figure supplement 1; Figure 5—figure supplement 1; Figure 5—figure supplement 2).

4) The authors state: "cold exposed Adpn-PRDM16 mice showed normal beiging, as expected from previous studies (Figure 5—figure supplement 1) (Berry et al., 2016)". However, there are no data from PRDM16 KO mice in Berry et al. (2016). In addition, Cohen et al. (2014) showed that Adpn-PRDM16 mice are defective in both cold- and CL^-^induced browning in iWAT. The authors need to clarify how their work fits into the scheme of what's been published previously.

We apologize for the ambiguity and have removed this portion of the sentence. What we meant to emphasize in the original manuscript was that cold exposed Adpn-PRDM16 mice showed normal beiging, which was consistent with our previous Adiponectin-RFP beige adipocyte fate mapping studies which demonstrated negative tracing (Berry et al., 2016).

We thank the reviewers for noting the differences between our studies and several of our colleagues. The major difference being: straight Cre vs. tamoxifen inducible Cre^ERT2^. Cohen and colleagues used a global Adiponectin-Cre model to drive PRDM16 deletion thus it is not surprising that beiging was altered under both cold and CL316,243 conditions because all beige and white (new and pre-existing) adipocytes will express Adiponectin either in the mature state or as they differentiate into mature lipid+ cells (Wang eta l 2013; Jeffery E et al. 2014, Shao M et al., 2016). Therefore, PRDM16 would be deleted under any setting in which “new” adipocytes (white or beige) or interconverting adipocytes would be created. In our studies only pre-existing beige and white adipocytes, not progenitors or newly differentiating adipocytes, will be deleted in PRDM16 thus new cells are unaffected by the deletion.

This clarification has been added to the text. Thank you for pointing this out as it is an opportunity to highlight the differences between our studies and previous reports.

5) Conflicting data have been reported for beiging in Adrb3 KO mice (Jimenez, Eur. J Biochem. 2003, Barbatelli, AJPEM, 2010, de Jong, AJPEM, 2017). The authors should address how their data contribute to the previous controversy.

We thank the reviewers for highlighting this controversy and we have added a discussion point around it.

6) Blocking ADRB1 by IP injection leads to systemic effects, especially profound effects on cardiac function. Could any of the observations on beiging be secondary to other physiological changes?

We agree with the reviewer that we cannot rule out the potential systemic effects of talinolol on other tissues and organs. Talinolol is hydrophilic and has difficulty crossing the blood brain barrier (Neil-Dwyer et al. 1981 and Sourgens et al. 2003). The reviewers are correct the literature indicates that ADRB1 inhibition may have a role in blood pressure. However, in our studies, the drug was not administered during cold or CL316,243 exposure rather we administered the drug for 5 days at RT with a 24-hour window prior to administering the beiging stimuli. The half-life of talinolol is 6-7 hours thus the effects of heart rate should be minimal at the time of cold or CL316,243 exposure (1-day post treatment or ~3 half-life’s). But the effects are sufficient to alter the engagement of SMA+ mural cells to form beige adipocytes. Ongoing studies in the lab are focused on expanding and exploring the mechanism by which ADRB1 elicits its effects on SMA+ cells.

7) The title of the manuscript covers both cold- and ADRB3-induced beige cell formation, however the majority of the studies focused on identifying the origin of ADRB3-induced beige adipocytes. There is minimal data presented on the progenitor of cold-induced beige adipocytes. The title could be modified and narrowed to better match the scope of this study.

We have modified the title to better reflect the proposed studies. “Distinct cellular and molecular mechanisms of β3 adrenergic receptor induced beige adipocyte formation”.